# Design and Anticancer Properties of New Water-Soluble Ruthenium–Cyclopentadienyl Complexes

**DOI:** 10.3390/ph15070862

**Published:** 2022-07-14

**Authors:** Tânia S. Morais, Fernanda Marques, Paulo J. Amorim Madeira, Maria Paula Robalo, Maria Helena Garcia

**Affiliations:** 1Centro de Química Estrutural, Institute of Molecular Sciences, Faculdade de Ciências, Universidade de Lisboa, Campo Grande, 1749-016 Lisboa, Portugal; 2Centro de Ciências e Tecnologias Nucleares, Instituto Superior Técnico, Universidade de Lisboa, Estrada Nacional 10 (km 139,7), 2695-066 Lisboa, Portugal; fmarujo@ctn.tecnico.ulisboa.pt; 3Ascenza Agro, SA, Av. do Rio Tejo-Parq. Ind. Sapec Bay, 2910-440 Setúbal, Portugal; paulo.madeira@ascenza.rovensa.com; 4Centro de Química Estrutural, Institute of Molecular Sciences, Instituto Superior Técnico, Universidade de Lisboa, Av. Rovisco Pais, 1049-001 Lisboa, Portugal; mprobalo@deq.isel.ipl.pt

**Keywords:** water-soluble, ruthenium, cyclopentadienyl, anticancer, albumin

## Abstract

Ruthenium complexes are emerging as one of the most promising classes of complexes for cancer therapy. However, their limited aqueous solubility may be the major limitation to their potential clinical application. In view and to contribute to the progress of this field, eight new water-soluble Ru(II) organometallic complexes of general formula [RuCp(*m*TPPMS)_n_(L)] [CF_3_SO_3_], where *m*TPPMS = diphenylphosphane-benzene-3-sulfonate, for *n* = 2, L is an imidazole-based ligand (imidazole, 1-benzylimidazole, 1-butylimidazole, (1-(3-aminopropyl)imidazole), and (1-(4-methoxyphenyl)imidazole)), and for *n* = 1, L is a bidentate heteroaromatic ligand (2-benzoylpyridine, (di(2-pyridyl)ketone), and (1,2-(2-pyridyl)benzo-[b]thiophene)) were synthesized and characterized. The new complexes were fully characterized by NMR, FT-IR, UV–vis., ESI-HRMS, and cyclic voltammetry, which confirmed all the proposed molecular structures. The antiproliferative potential of the new Ru(II) complexes was evaluated on MDAMB231 breast adenocarcinoma, A2780 ovarian carcinoma, and HT29 colorectal adenocarcinoma cell lines, showing micromolar (MDAMB231 and HT29) and submicromolar (A2780) IC_50_ values. The interaction of complex **6** with human serum albumin (HSA) and fatty-acid-free human serum albumin (HSA^faf^) was evaluated by fluorescence spectroscopy techniques, and the results revealed that the ruthenium complex strongly quenches the intrinsic fluorescence of albumin in both cases.

## 1. Introduction

Cancer remains one of the major health problems worldwide [1]. The clinical success of platinum anticancer agents, which are still used as first-line anticancer drugs, has fostered an increasing research interest in metallodrugs [2,3,4,5,6,7,8,9]. Therefore, several other transition metal complexes have been developed to prevent, treat, and diagnose malignant cancers. In this regard, ruthenium complexes have emerged as one of the leading candidates for alternative platinum drugs [2,3,9,10,11,12]. In fact, ruthenium complexes are the second most-studied class of anticancer metal complexes due to their inherent advantages when compared with platinum-based ones, including multiple stable oxidation states, a higher number of coordination positions available, alternative coordination geometries, low ligand exchange rates with kinetics on the timescale of cell processes, high cytotoxicity, lower side effects, and different mechanisms of action [11,13,14,15]. Among them, some ruthenium compounds have revealed outstanding in vivo and in vitro activities, namely NAMI-A(Imidazolium-[*trans*-tetrachloro(dimethylsulfoxide)imidazoleruthenium(III)]) [16,17], KP1019(indazolium[*trans*-[tetrachlorobis(indazole)ruthenate(III)]) [18,19], NKP-1339 (the sodium salt of KP1019, sodium trans-[tetrachloridobis(1H-indazole)ruthenate(III)]) [20,21,22], and TLD1433 ([RuII(4,4′-dimethyl-2,2′-bipyridine)2(2-(2′,2″:5″,2‴-terthiophene)-imidazol [4,5-f] [1,10] phenathroline)] (Figure 1) [23]. Nevertheless, only complexes NKP-1339 and NAMI-A, both developed as chemotherapeutic agents, and TLD1433, developed as a photosensitizer for photodynamic therapy, have reached phase I/II of clinical trials, in spite of significant efforts [24].

Organometallic complexes have also attracted much attention as effective anticancer agents. A large group of organometallic Ru(II)-arene complexes have been reported as promising potential metallodrugs [25,26,27,28,29,30]. The most-studied organoruthenium compounds, ruthenium(II) *p*-cymene derivatives in combination with mono-/bidentate ligands and halide coordination, have proven to be potent cytotoxic agents against a range of tumor cell lines in vitro [12,31], showing in some cases antimetastatic and antiangiogenic behaviors in vivo [32,33]. In this frame, we have been engaged in the development of half-sandwich organometallic complexes based on the ruthenium(II)-cyclopentadienyl (RuCp) fragment as prospective anticancer agents [34,35,36,37,38,39]. Given the promising in vitro and in vivo antitumor properties obtained for our lead RuCp(2,2′bipy)(PPh_3_)][CF_3_SO_3_] (TM34) [34,35,36], and being aware that the limited aqueous solubility of metallodrugs is one of the major limitations to its possible clinical application, we designed and synthesized a new molecular structure [RuCp(2,2′bipy)(*m*TPPMSNa)][CF_3_SO_3_](TM85) analogous to our lead TM34 that showed interesting in vitro and in vivo antitumor properties [38,40]. Nevertheless, TM85 gained in solubility when compared to TM34 but showed a significant decrease in activity, in particular in more aggressive cell lines. This observation clearly corroborates the well-established finding that small structural changes produce large variations in the mode of action at the cellular level [13].

In order to further explore the chemical and biological properties of RuCp water-soluble compounds, we report here the synthesis and characterization of eight new complexes of general formula [RuCp(*m*TPPMS)*_n_*(L)][CF_3_SO_3_], where *m*TPPMS = diphenylphosphane-benzene-3-sulfonate, for *n* = 2, L is an imidazole-based ligand (imidazole, 1-benzylimidazole, 1-butylimidazole, (1-(3-aminopropyl)imidazole), and (1-(4-methoxyphenyl) imidazole)), and for *n* = 1 L, is a bidentate heteroaromatic ligand (2-benzoylpyridine, di(2-pyridyl)ketone, and 1,2-(2-pyridyl)benzo-[b]thiophene). The choice of these ligands was based on the promising results previously achieved for analogous complexes containing the {RuCp(PPh_3_)} fragment [36]. The antiproliferative activity of Ru(II)Cp complexes was assessed in three tumor cell lines. For the most promising complex, as an initial approach to outline its pharmacokinetics, its interaction with human serum albumin as a vehicle for transport in blood plasma was also investigated.

## 2. Results and Discussion

Eight new water-soluble Ru^II^Cp complexes of general formula [RuCp(*m*TPPMS)*_n_*(L)][CF_3_SO_3_] were prepared in high yields (84–92%) by halide abstraction with AgCF_3_SO_3_ from the precursor [RuCp(*m*TPPMS)_n_Cl], where, for *n* = 2, L is an imidazole-based ligand (imidazole, 1-benzylimidazole, 1-butylimidazole, 1-(3-aminopropyl)imidazole, and 1-(4-methoxyphenyl)imidazole), and for *n* = 1, L is a bidentate heteroaromatic ligand (2-benzoylpyridine, (di(2-pyridyl)ketone), and (1,2-(2-pyridyl)benzo-[b]thiophene)). The reactions were carried out in methanol solutions at reflux or stirring at room temperature (Figure 1). The proposed structural details were elucidated by FT-IR, ^1^H, ^13^C and ^31^P NMR, UV–vis spectroscopies, cyclic voltammetry, and accurate mass measurements.

### 2.1. NMR, FT-IR Analysis, and Mass Spectral Analysis

NMR characterization of the complexes was carried out by ^1^H, ^13^C{^1^H}, ^31^P{^1^H}, and 2D experiments (COSY, HSQC, and HMBC). Figure 1 shows the numbering of the coordinated heteroaromatic ligands for simplicity of the present discussion.

After coordination of the heteroaromatic ligands a deshielding on the Cp, up to 0.45 ppm, is observed, as expected for monocationic RuCp complexes. For complexes **1**–**5**, the effect of coordination of imidazole-based ligands is observed by a deshielding of the protons adjacent to the coordinated nitrogen atom and a shielding of the remaining protons. This effect is probably due to an influence of the organometallic fragment on the electronic flow towards the heteroaromatic ring, which already been observed for other Ru(II) piano-stool complexes with nitrogen-coordinated heteroaromatic ligands such as pyridylpyrazoles, pyridylimidazoles, and phenoxazine [27,37,40,41]. The effect of the coordination of the N,O- bidentate ligand in compound **6** is characterized by a significant deshielding of the H1 proton of the bopy ligand. This behavior has already been observed for the analogous complex [Ru(η^5^-C_5_H_5_)(PPh_3_)(bopy)][CF_3_SO_3_] [37], and it is in accordance with a purely sigma coordination by the nitrogen atom. For the protons closer to oxygen, a significant shielding is observed, suggesting that the π-backdonation for the ligand takes place through the coordinated O atom. For compound **7**, the effect of the coordination of the di(2-pyridyl)ketone ligand through the two nitrogen atoms, which is remarkable for H1 and H2 protons, is a consequence of an electronic flow towards the aromatic ligand due to π-backdonation involving the d orbital of the ruthenium center and the π* orbital located in the nitrogen atoms. In complex **8**, after coordination of the 2-(2-pyridyl)benzo[b]thiophene (pbt) ligand, deshielding of the H1 proton is observed, characteristic of sigma coordination by the nitrogen atom. It is not possible to make an analysis of the protons adjacent to the coordinated S atom since the protons are overlapped by the phosphine ligands. ^13^C NMR spectra revealed the same general effect observed for the protons. ^31^P NMR spectra are characterized by a single sharp signal for the *m*TPPMS phosphine coligand in the range of 43 to 54 ppm by expected deshielding upon coordination according to the σ donor character of this phosphine (Δ ≈ 50 ppm).

The solid-state FT-IR spectra of all complexes present the characteristic bands of the cyclopentadienyl rings (ν_C-H_ ≈ 3050 cm^−1^) and the phenyl aromatic rings (ν_C-H_ at 3040–3100 cm^−1^; ν_C=C_ ≈ 1435 cm^−1^), the counter-anion (ν(CF_3_SO_3_) at 1250–1267 cm^−1^), and the sulfonate group (ν_S=O_1195 cm^−1^). For compound **6**, the coordination of the ketonic functional group to the ruthenium center leads to the displacement of ν_C=O_ at ≈ 1700 cm^−1^ to lower energy, while for compound **7**, the ν_C=O_ vibration appears at 1600 cm^−1^, proving the preference of ruthenium for *N,N′* chelation. The accurate mass ESI-HRMS measurements were in accordance with the proposed formulation, confirming the formation of the complexes.

### 2.2. Electronic Absorption Spectroscopy

The electronic spectra of complexes [Ru(η^5^-C_5_H_5_)(*m*TPPMS)*_n_*(L)][CF_3_SO_3_] (**1**–**8**) were recorded in 10^−4^ to 10^−6^ mol dm^−3^ solutions of methanol and water (see Table 1). Figure 2 presents the typical electronic spectra of this family of compounds. The spectra of all the complexes displayed strong absorption bands in the range of 225–330 nm of the UV region, attributed to π-π* electronic transitions occurring at the {Ru(η^5^-C_5_H_5_)(*m*TPPMS)*_n_*}^+^ organometallic fragment and the coordinated heteroaromatic ligands. In addition, the electronic spectra of compounds with bidentate ligands (**6**–**8**) in methanol displayed one or two less intense maximum absorptions between 420 and 540 nm, characteristic of the metal–ligand charge transfer (MLCT) bands involving the π*orbital of the heteroaromatic ligands and ruthenium 4d orbitals. The imidazole-based complexes (**1**–**5**) do not present any band in this region. For a complete characterization, electronic spectra of these compounds were also obtained in water solutions (Table 1). The charge transfer bands for compounds **6**–**8** appear at slightly higher energy values, as expected, since water is more polar than methanol but with lower intensities.

### 2.3. Electrochemical Characterization of Complexes

The electrochemical study by cyclic voltammetry (CV) of complexes [Ru(η^5^-C_5_H_5_)(*m*TPPMSNa)_2_(L)][CF_3_SO_3_] (**1**–**5**) and [Ru(η^5^-C_5_H_5_)(*m*TPPMSNa)(L)][CF_3_SO_3_] (**6**–**8**), was run at room temperature in dichloromethane and aqueous HEPES (10 mM, pH 7.4) buffer media, and the results are summarized in Table 2. Due to the insolubility of complexes **1**–**5** in dichloromethane, these were only conducted in HEPES buffer solution. The electrochemical behavior of the ligands was also studied in both solvents, revealing redox inactivity within the potential window used. Figure 3 shows the cyclic voltammograms of a representative example of the bidentate complexes recorded in 0.2 M [*n*-Bu_4_N][PF_6_]/dichloromethane and HEPES-buffered 7.4 solutions. The other cyclic voltammograms are presented in the Appendix A.

Complexes [Ru(η^5^-C_5_H_5_)(*m*TPPMSNa)(L)][CF_3_SO_3_] (**6**–**8**) (L = bopy, dpk, and pbt) are redox-active in dichloromethane, showing for the ruthenium(II/III) couple an irreversible oxidation process in the range of 1.02 V–1.17 V (see Table 2). When the scan direction is reversed just after the oxidation process, a very weak cathodic wave is observed. This redox behavior is in agreement with the one reported before for complex [Ru(η^5^-C_5_H_5_)(*m*TPPMSNa)(2,2′-bipy)][CF_3_SO_3_] [40] and indicative of an unstable Ru(III) species formation, followed by fast decomposition. The E_pa_ values found for complexes 6–8 are slightly more anodic than those observed for the related complexes [Ru(η^5^-C_5_H_5_)(PPh_3_)(L)][CF_3_SO_3_] (L = bopy, dpk, and pbt) [37,40], for which the oxidation potentials are, respectively, 0.99 V, 0.98 V, and 1.11 V. Furthermore, in contrast to the triphenylphosphine derivatives for which ligand-based processes were observed when scanning in the cathodic direction, in this case, no reduction processes were observed. This indicates that the presence of the water-soluble phosphine coligand influences the electronic environment around the ruthenium(II) center, making the metal-centered oxidation and the bidentate ligand reduction processes more difficult.

The water solubility of the complexes under study and the interest in electrochemical studies in aqueous (buffer) media, closer to the conditions found in biological media, when dealing with potential anticancer drug candidates led to the study of complexes **1**–**8** in 100 mM HEPES buffer (pH 7.4 at v = 50 mV·s^−1^). The general behavior of all complexes is described by oxidation of ruthenium(II/III) in the range of 0.77 V–1.17 V, and a representative example, complex **6**, is shown in Figure 3. The introduction of aliphatic or aromatic chains at the nitrogen atom of the imdazole ligand leads to slightly more positive oxidation potentials for the ruthenium(II) center, whereas structural modifications in the alkyl chain do not seem to influence these potentials. Solvent exchange for complexes **6**–**8** induces a shift to lower potentials up to 330 mV for ruthenium(II/III) oxidation (see Table 2), which indicates a decrease in the stability at the ruthenium(II) center in aqueous media.

### 2.4. Complex Solubility and Stability in Aqueous Solutions and Reactivity towards O_2_ and Estimation of Lipophilicity

All the complexes are highly soluble in methanol and in water (except complex **7** in methanol). The solubility of these complexes in water can be anticipated due to the presence of one or two water-soluble phosphine ligands coordinated to the {Ru(η^5^-C_5_H_5_)} fragment. Table 3 summarizes the water-solubility values found for all the complexes. Generally, there is a dependence between the solubility of the complex and the number of coordinated phosphine ligands, except for complexes **1** and **5**, for which the heteroaromatic ligand probably confers some insolubility.

The stability of the complexes in water was evaluated by UV–vis and ^1^H and ^31^P NMR spectroscopies (Table 3). As expected, compounds with bidentate heteroaromatic ligands (**6**–**8**) are much more stable than compounds with monodentate ligands. Changes observed in the UV–vis spectra of complexes **6** and **7** over 4 days at 37 °C were insignificant, indicating that these complexes are air-stable in aqueous solution. These results were supported by NMR spectroscopy; in the ^31^P NMR spectra, no changes in the number of peaks displayed or in their chemical shift (δ) values were observed over 4 days, in D_2_O. The UV–vis spectrum of compound **8** shows a decrease in the intensity of the bands due to precipitation; however, it is possible to conclude by NMR that this compound remained intact without the formation of new species. Figure 4 represents the evaluation of the stability on the charge transfer of complex **6** by UV–vis and ^31^P-NMR spectroscopies. Complexes with imidazole-based ligands (**1**–**5**) have been shown to undergo hydrolysis after a few hours.

The determination of hydrophobicity/lipophilicity of the compounds is a key issue in the development of new drugs since it affects their absorption, blood–brain distribution, drug–receptor interaction, etc. [42]. The lipophilicity of a drug can affect its tissue permeability, which can influence its localization in the target tissues and its capacity to binding to biomolecules. The n-octanol/water partition coefficient was measured using the shake-flask method at room temperature. The log P_o/w_ values were determined only for compounds **4**, **6**, and **7**; for the remaining compounds, they were not determined due to their instability and/or precipitation in water. The estimated log P_o/w_ values (Table 3) indicate that all the studied compounds are very lipophilic. Indeed, the presence of two water-soluble phosphine ligands in compound **4** makes it less lipophilic than complexes **6** and **7**.

### 2.5. Cytotoxicity in Human Cell Lines

The cytotoxic activity of [Ru(η^5^-C_5_H_5_)(*m*TPPMS)(L)] [CF_3_SO_3_] complexes (**6**–**8**) was assessed in the ovarian A2780, triple-negative breast MDAMB231, and colon HT29 cancer cells lines by the colorimetric MTT assay after 72 h exposure to different concentrations of complexes **6**–**8**. The cytotoxicity of the remaining compounds was not determined due to the instability of the complexes.

This method allows the evaluation of the cellular viability of each complex for the studied cell lines as an indicator of the anticancer efficiency of the complexes. All the complexes exhibited high to moderate cytotoxicity against the three cell lines, being much more active in the A2780 cells (Table 4) with the exception of complex **7**, which is not cytotoxic for the MDAMB231 and HT29 cell lines. None of the heteroaromatic organic ligands were cytotoxic up to 100 μM. Besides the clear selectivity revealed for the A2780 cell line, these results also suggest that there is no relationship between cytotoxicity and the number of phosphines.

When comparing these results with the non-water-soluble–related RuCp complexes [34,35,36,37], it is possible to conclude that the introduction of sulfonated phosphine ligands leads to an increase in water solubility of the complexes but slightly decreases their cytotoxic potential, in particular for the most chemoresistant MDAMB231 and HT29 tumor cells. However, the cytotoxic activity does not differ for the cisplatin-sensitive A2780 cells.

### 2.6. Fluorescence Quenching of HSA by Complex [Ru(η^5^-C_5_H_5_)(mTPPMS)(bopy)][CF_3_SO_3_] (**6**)

As ruthenium complexes are reported to bind albumin, we investigated the interaction between [Ru(η^5^-C_5_H_5_)(*m*TPPMS)(bopy)][CF_3_SO_3_] (**6**) and human serum albumin (HSA) by fluorescence spectroscopy. Complex **6** was chosen for being the most promising in terms of solubility, stability, and cytotoxicity. Fluorescence quenching occurs when a drug approaches the excited fluorophore as a consequence of energy transfer or electron transfer between fluorophore and drug. The HSA exhibits intrinsic fluorescence due to the contribution of three fluorophores: tyrosine, tryptophan, and phenylalanine amino acid residues [46]. However, the main contribution comes from the tryptophan residue due to the low-quantum phenylalanine and tyrosine yields [46,47]. Since HSA has only one tryptophan residue, Trp214, that can be selectively excited at 295 nm, quenching of its intrinsic fluorescence was employed as a probe atao evaluate the interaction between complex **6** and HSA. Figure 5 shows the HSA and HSA^faf^ fluorescence spectra obtained in the absence and the presence of increasing ruthenium complex concentrations (0–45 μM) at near-physiological conditions, at pH 7.4, after incubation at 298 K for 24 h. HSA and HSA^faf^ show an emission maximum at 335 nm, and the addition of complex **6** to HSA and HSA^faf^ resulted in a gradual decrease in the fluorescence intensity, leading to 75% and 97% of quenching at the highest Ru concentration, respectively. These results clearly indicated the binding of the Ru complex to HSA, changing the microenvironment around Trp-214 residue and the tertiary structure of albumin. Although the presence of fatty acids in protein does not hinder the binding of complex **6** to albumin, stronger binding is observed in the absence of fatty acids, which indicates that complex **6** can also occupy the binding sites on HSA for fatty acids. In addition, an isosbestic point was observed around 430 nm in the HSA/HSA^faf^ -Ru systems, which might also prove the formation of {Ru-HSA} and {Ru-HSA^faf^} fluorescent complexes.

Fluorescence quenching can occur by distinct mechanisms that require molecular contact between the quencher and the fluorophore, typically classified as static or dynamic. Dynamic quenching refers to collisions between the fluorophore and the quencher during the transient existence of the excited state, without any permanent change in both molecules, while static quenching refers to the formation of a fluorophore–quencher complex [46,48]. To investigate the quenching mechanism of HSA with the Ru complex, fluorescence quenching data were analyzed using the Stern−Volmer equation (Equation (S1), Supporting Information). From the plots of F_0_/F versus [Q] (Appendix A, Supporting Information), the values of K_SV_ and K_q_ were calculated and are listed in Table 5. These results point out good linearity at the three different temperatures, and the increase with the temperature of Stern–Volmer quenching constants KSV suggests a dynamic fluorescence quenching process. Nonetheless, the K_q_ values of the HSA-Ru and HSA^faf^-Ru systems at the three temperatures are higher than the maximum diffusion collision quenching rate constant (2.0 × 10^10^ L^−1^ mol^−1^ s^−1^), revealing that some static quenching mechanism should also be involved in these interaction systems. Thus, the fluorescence quenching is initiated by a combined process (dynamic and static mechanisms). The K_SV_ values found for this compound in both HSA variants are higher than those found for the non-water-soluble–related complex [Ru(η^5^-C_5_H_5_)(PPh_3_)(bopy)][CF_3_SO_3_] (TM90) [49], thus suggesting a higher affinity to albumin.

Equation (S2) was employed to obtain the association constant for a site (K_a_) and the number of binding sites from the emission spectra. The binding equilibrium plots for the fluorescence quenching of HSA and HSA^faf^ by [Ru(η^5^-C_5_H_5_)(*m*TPPMS)(bopy)][CF_3_SO_3_] (**6**) are shown in Appendix A, and the values of K_a_, *n* and correlation coefficients are listed in Table 6. The *n* values for these systems were approximately equal to 1, indicating the existence of a single binding site in the two variants of HSA that is reactive to the Ru complex. The K_a_ values decrease with the temperature rising, which may indicate the formation of an unstable compound that partially decomposes at higher temperatures [50].

The noncovalent interactions between a protein and a drug include multiple hydrogen bonds, hydrophobic interactions, and van der Waals and electrostatic forces [49]. The major binding forces between a drug and a protein can be estimated through the magnitude and sign of the enthalpy change (ΔH), entropy change (ΔS), and free energy change (ΔG). To better understand the interaction between the HSA and HSA^faf^ with Ru complex **6**, the thermodynamic parameters were calculated using Van’t Hoff equations (Equations (S3) and (S4)) and plots (Appendix A). From Table 6, it can be observed that the negative values of ΔG indicate that the interaction of the Ru complex with HSA and HSA^faf^ is spontaneous. Both positive ΔS and ΔH suggest that hydrophobic interactions are the major forces between the Ru complex and albumin.

To confirm the binding of the [Ru(η^5^-C_5_H_5_)(*m*TPPMS)(bopy)][CF_3_SO_3_] (**6**) complex to site I (also called warfarin site) of HSA, competitive studies between the ruthenium compound and warfarin (a site marker for Sudlow site I) were carried out. In this way, the Ru complex was added to the solution of {HSA-warfarin} adduct (emits intensively at ~380 nm when excited at 305 nm). As seen in Figure 6, the fluorescence intensity of the {HSA-warfarin} adduct decreases upon increasing the concentration of the Ru complex, suggesting that warfarin had been partially replaced by this complex. Therefore, these results suggest that the binding site of this Ru complex on HSA can be proposed as the same as the warfarin binding site or Sudlow site I.

## 3. Materials and Methods

### 3.1. Materials and General Procedures

Starting reagents and solvents were used as received from standard chemical suppliers, unless otherwise stated. The doubly purified water used in all experiments was from a Millipore^®^ system. All organometallic syntheses were carried out under dinitrogen atmosphere using current Schlenk techniques. The solvents used were previously distilled under nitrogen atmosphere before use, according to standard literature methods. Starting material [RuCp(*m*TPPMSNa)_2_Cl] was prepared following the literature procedure [51]. The NMR spectra were recorded on a Bruker Avance 400 spectrometer (^1^H NMR at 400.13 MHz, ^13^C NMR at 100.6 MHz, ^31^P NMR at 161.97 MHz) at probe temperature. ^1^H and ^13^C chemical shifts were reported downfield relative to solvent peaks considering internal Me_4_Si (0 ppm), and ^31^P NMR chemical shifts were reported downfield relative to externally referenced 85% H_3_PO_4_. Chemical shifts (δ) are reported in parts per million (ppm), and the resonance multiplicity is expressed as singlet (s), doublet (d), triplet (t), quintet (quint), sextet (sext), multiplet (m), and complex (comp). All the assignments were attributed using COSY, HSQC, and HMBC 2D-RMN techniques. Samples were prepared under air and at room temperature, using methanol deuterated solvent. The infrared spectra (4000–400 cm^−1^) were recorded by using a Thermo Nicolet 6700 spectrophotometer in dry KBr pellets, with only significant bands being cited in the text. ESI-HRMS (HRMS = high-resolution mass spectrometry) spectra were acquired in an Apex Ultra FTICR Mass Spectrometer equipped with an Apollo II Dual ESI/MALDI (electrospray ionization/matrix-assisted laser desorption/ionization) ion source, from Bruker Daltonics, and a 7 T actively shielded magnet from Magnex Scientific. Electronic UV–visible spectra were recorded at room temperature, using 1 cm optical path quartz cells, on a Jasco V-660 spectrometer in the range of 220–900 nm. Albumin and warfarin samples were purchased from Sigma-Aldrich; fatty acid HSA (≥96% lyophilized powder, A1653), fatty acid-free HSA (approx. 99%, lyophilized powder, A3782), and warfarin (A2250) were used as received.

### 3.2. Chemical Synthesis

#### 3.2.1. General Procedure for the Synthesis of [RuCp(mTPPMSNa)_2_L][CF_3_SO_3_] Complexes (**1**–**5**)

To a stirred solution of [RuCp(*m*TPPMSNa)_2_Cl] (0.5 mmol) in methanol (25 mL) was added AgCF_3_SO_3_ (0.5 mmol) and the respective imidazole (0.5 mmol). The reaction was followed for 18 to 20 h at room temperature and monitored by ^1^H and ^31^P NMR. The solution was separated from the AgCl precipitate by cannula-filtration and the solvent evaporated under vacuum. The product was washed with *n*-hexane (3 × 10 mL) and recrystallized from dichloromethane/diethyl ether.

Data for [RuCp(*m*TPPMSNa)_2_(ImH)][CF_3_SO_3_] (**1**): Yellow powder; yield: 88%.

^1^H NMR [CD_3_OD, Me_4_Si, δ/ppm]: 8.11 (s, 1, H1), 7.87 (d, 1, H2), 7.64–6.92 (m, 29, *m*TPPMS + H3), 4.50 (s, 5, η^5^-C_5_H_5_). ^13^C{^1^H} NMR [CD_3_OD, δ/ppm]: 146.43 (C1), 128.42 (C2), 141.24–118.69 (singlets of aromatic *m*TPPMS + C3), 83.72 (η^5^-C_5_H_5_). ^31^P NMR [CD_3_OD, δ/ppm]: 43.01 (s, *m*TPPMS). FT-IR [KBr, cm^−1^]: 3100–3040 cm^−1^ (ν_C-H_, Cp, imidazole and phenyl rings), 1435 cm^−1^ (ν_C=C_, phenyl rings), 1260 cm^−1^ (ν(CF_3_SO_3_)), 1195 cm^−1^ (ν(SO_3_)). ESI-HRMS: calc. for [M^+^] 957.043415, found 957.04352.

Data for [RuCp(*m*TPPMSNa)_2_(1-BI)][CF_3_SO_3_] (**2**): Yellow powder; yield: 89%.

^1^H NMR [CD_3_OD, Me_4_Si, δ/ppm]: 8.12 (s, 1, H1), 7.93 (d, 1, H2), 7.43–6.85 (comp, 35, mTPPMS + H3 + H5 + H6 + H7 + H8 + H9 + H10), 4.68 (s, 2, H4), 4.49 (s, 5, η^5^-C_5_H_5_). ^13^C{^1^H} NMR [CD_3_OD, δ/ppm]: 146.60 (C1), 143.19 (C5), 128.80 (C2), 136.74–122.91 (singlets of aromatic *m*TPPMS + C3 + C6 + C7 + C8 + C9 + C10), 84.02 (η^5^-C_5_H_5_), 52.71 (C4). ^31^P NMR [CD_3_OD, δ/ppm]: 43.67 (s, *m*TPPMS). FT-IR [KBr, cm^−1^]: 3100–3040 cm^−1^ (ν_C-H_, Cp, imidazole and phenyl rings), 1435 cm^−1^ (ν_C=C_, phenyl rings), 1263 cm^−1^ (ν(CF_3_SO_3_)), 1195 cm^−1^ (ν(SO_3_)). ESI-HRMS: calc. for [M^+^] 1047.090365, found 1047.09161.

Data for [RuCp(*m*TPPMSNa)_2_(1-BuIm)][CF_3_SO_3_] (**3**): Yellow powder; yield: 92%.

^1^H NMR [CD_3_OD, Me_4_Si, δ/ppm]: 8.12 (s, 1, H1), 7.89 (d, 1, H2), 7.48–6.72 (m, 29, *m*TPPMS + H3), 4.51 (s, 5, η^5^-C_5_H_5_), 3.56 (t, 2, H4), 1.35 (quint, 2, H5), 0.88 (sext, 2, H6), 0.77 (t, 3, H7). ^13^C{^1^H} NMR [CD_3_OD, δ/ppm]: 146.63 (C1), 128.67 (C2), 136.66–122.02 (singlets of aromatic *m*TPPMS + C3), 83.96 (η^5^-C_5_H_5_), 48.36 (C4), 33.43 (C5), 20.32 (C6), 13.89 (C7). ^31^P NMR [CD_3_OD, δ/ppm]: 43.35 (s, *m*TPPMS). FT-IR [KBr, cm^−1^]: 3100–3040 cm^−1^ (ν_C-H_, Cp, imidazole and phenyl rings), 1436 cm^−1^ (ν_C=C_, phenyl rings), 1260 cm^−1^ (ν(CF_3_SO_3_)), 1195 cm^−1^ (ν(SO_3_)). HRMS: calc. for [M^+^] 1013.106015, found 1013.10614.

Data for [RuCp(*m*TPPMSNa)_2_(3-ApIm)][CF_3_SO_3_] (**4**): Yellow powder; yield: 84%.

^1^H NMR [CD_3_OD, Me_4_Si, δ/ppm]: 8.08 (s, 1, H1), 7.98 (d, 1, H2), 7.68–6.89 (m, 29, *m*TPPMS + H3), 4.89 (H7 + MeOD), 4.49 (s, 5, η^5^-C_5_H_5_), 3.69 (t, 2, H4), 2.34 (quint, 2, H5), 1.61 (quint, 2, H6). ^13^C{^1^H} NMR [CD_3_OD, δ/ppm]: 146.70 (C1), 128.42 (C2), 138.50–120.80 (singlets of aromatic *m*TPPMS + C3), 84.05 (η^5^-C_5_H_5_), 46.36 (C4), 39.23 (C5), 33.82 (C6). ^31^P NMR [CD_3_OD, δ/ppm]: 43.16 (s, *m*TPPMS). FT-IR [KBr, cm^−1^]: 3100–3040 cm^−1^ (ν_C-H_, Cp, imidazole and phenyl rings), 1436 cm^−1^ (ν_C=C_, phenyl rings), 1265 cm^−1^ (ν(CF_3_SO_3_)), 1195 cm^−1^ (ν(SO_3_)). ESI-HRMS: calc. for [M^+^] 1020.09913, found 1020.10106.

Data for [RuCp(*m*TPPMSNa)_2_(4-MpIm)][CF_3_SO_3_] (**5**): Yellow powder; yield: 90%.

^1^H NMR [CD_3_OD, Me_4_Si, δ/ppm]: 8.23 (s, 1, H1), 7.90 (d, 1, H2), 7.48–6.90 (m, 33, *m*TPPMS + H3 + H5 + H6 + H8 + H9), 4.58 (s, 5, η^5^-C_5_H_5_), 3.81 (s, 3, H10). ^13^C{^1^H} NMR [CD_3_OD, δ/ppm]: 161.05 (C7), 146.81 (C1), 130.40 (C4), 128.73 (C2), 135. 12–115.82 (singlets of aromatic *m*TPPMS + C3 + C5 + C6 + C8 + C9), 84.02 (η^5^-C_5_H_5_), 5608 (C10). ^31^P NMR [CD_3_OD, δ/ppm]: 43.27 (s, *m*TPPMS). FT-IR [KBr, cm^−1^]: 3100–3040 cm^−1^ (ν_C-H_, Cp, imidazole and phenyl rings), 1435 cm^−1^ (ν_C=C_, phenyl rings), 1252 cm^−1^ (ν(CF_3_SO_3_)), 1195 cm^−1^ (ν(SO_3_)). ESI-HRMS: calc. for [M^+^] 1063.085280, found 1063.08806.

#### 3.2.2. General Procedure for the Synthesis of [RuCp(mTPPMSNa)L][CF3SO3] Complexes (**6**–**8**)

To a stirred solution of [RuCp(*m*TPPMSNa)_2_Cl] (0.5 mmol) in methanol (25 mL) was added AgCF_3_SO_3_ (0.5 mmol) and the respective bidentate ligand (0.5 mmol). After refluxing for 5 h, the solution was separated from the AgCl precipitate by cannula-filtration, and the solvent was evaporated under vacuum. The product was washed with *n*-hexane (3 × 10 mL) and recrystallized from dichloromethane/diethyl ether.

Data for [RuCp(*m*TPPMSNa)(bopy)][CF_3_SO_3_] (**6**): Purple powder; yield: 87%.

^1^H NMR [CD_3_OD, Me_4_Si, δ/ppm]: 9.71 (d, 1, H1), 8.25 (d, 1, H4), 7.16 (t, 1, H2), 7.05 (t, 1, H10), 7.93–7.31 (m, 19, *m*TPPMS + H3 + H8 + H9 + H11 + H12), 4.66 (s, 5, η^5^-C_5_H_5_). ^31^P NMR [CD_3_OD, δ/ppm]: 49.62 (s, *m*TPPMS). FT-IR [KBr, cm^−1^]: 3100–3040 cm^−1^ (ν_C-H_, Cp and phenyl rings), 1300–1400 cm^−1^ (ν_C=O_), 1433 cm^−1^ (ν_C=C_, phenyl rings), 1267 cm^−1^ (ν(CF_3_SO_3_)), 1195 cm^−1^ (ν(SO_3_)). ESI-HRMS: calc. for [M^+^] 714.4213, found 714.03997. This compound is not sufficiently soluble in methanol in sufficient quantities to perform 13C NMR experiment.

Data for [RuCp(mTPPMSNa)(dpk)][CF_3_SO_3_] (**7**): Red powder; yield: 88%.

^1^H NMR [CD_3_OD, Me_4_Si, δ/ppm]: 8.41 (d, 2, H1), 8.17 (d, 2, H4), 8.09 (m, 2, H3), 7.02 (m, 2, H2), 7.98–6.92 (m, 14, mTPPMS), 4.64 (s, 5, η^5^-C_5_H_5_). ^13^C{^1^H} NMR [CD_3_OD, δ/ppm]: 159.68 (C1), 157.38 (C6(C=O)), 150.14 (C5), 130.27 (C3), 129.0 (C4), 127.88 (C2), 139.31–123.41 (singlets of aromatic *m*TPPMS), 79.60 (η^5^-C_5_H_5_). ^31^P NMR [CD_3_OD, δ/ppm]: 53.67 (s, *m*TPPMS). FT-IR [KBr, cm^−1^]: 3100–3040 cm^−1^ (ν_C-H_, Cp and phenyl rings), 1435 cm^−1^ (ν_C=C_, phenyl rings), 1600 cm^−1^ (ν_C=O_), 1264 cm^−1^ (ν(CF_3_SO_3_)), 1195 cm^−1^ (ν(SO_3_)). ESI-HRMS: calc. for [M^+^] 715.03734, found 715.03605.

Data for [RuCp(*m*TPPMSNa)(pbt)][CF_3_SO_3_] (**8**): Red powder; yield: 91%.

^1^H NMR [CD_3_OD, Me_4_Si, δ/ppm]: 8.80 (d, 1, H1), 8.55 (d, 1, H4), 8.27 (d, 1, H12), 8.19 (d, 1, H9), 8.09 (t, 1, H11), 8.05–6.99 (m, 18, *m*TPPMS + H7 + H10 + H3 + H2), 4.52 (s, 5, η^5^-C_5_H_5_). ^13^C{^1^H} NMR [CD_3_OD, δ/ppm]: 154.50 (C1), 150.10 (C4), 132.98 (C9), 130.01 (C11), 129.41 (C13), 126.66 (C12), 123.58 (C8), 141.79–121.78 (singlets of aromatic *m*TPPMS + C2 + C3 + C5 + C6 + C7 + C10), 79.53 (η^5^-C_5_H_5_). ^31^P NMR [CD_3_OD, δ/ppm]: 50.41 (s, *m*TPPMS). FT-IR [KBr, cm^−1^]: 3100–3040 cm^−1^ (ν_C-H_, Cp and phenyl rings), 1435 cm^−1^ (ν_C=C_, phenyl rings), 1600 cm^−1^ (ν_C=O_), 1266 cm^−1^ (ν(CF_3_SO_3_)), 1195 cm^−1^ (ν(SO_3_)). ESI-HRMS: calc. for [M^+^] 742.01921, found 742.01648.

### 3.3. Cyclic Voltammetry

The cyclic voltammograms were obtained with an EG&G Princeton Applied Research Potentiostat/Galvanostat Model 273A equipped with Electrochemical PowerSuite v2.51 software in anhydrous dichloromethane (with 0.2 M of tetrabutylammonium hexafluorophosphate as supporting electrolyte) or HEPES (4-(2-hydroxyethyl)-1-piperazineethanesulfonic acid) buffer 100 mM, pH 7.4 solutions. For the organic solutions, a homemade three-electrode configuration cell with a platinum-disk working electrode (1.0 mm) probed by a Luggin capillary connected to a silver-wire pseudo-reference electrode and a platinum wire auxiliary electrode was used. The reported potentials were measured against the ferrocene/ferrocenium redox couple as internal standard and normally quoted relative to SCE (using the ferrocenium/ferrocene redox couple E1/2 = 0.46 V versus SCE). The electrochemical experimental window was −1.8 to 1.8 V, and the sweep rate range was 50–1000 mVs^−1^. For the HEPES buffer solution experiments, a glassy carbon (GC) disk electrode (3.0 mm) was used as the working electrode, the counter electrode was a platinum wire, and a saturated calomel electrode (SCE) was used as reference. The electrode GC surface was refreshed before each measurement by polishing with alumina, and the final cleaning was performed in water in an ultrasonic bath. The electrochemical measurements were performed in the −0.3 to 1.2 V potential range at a sweep rate of 50 mVs^−1^. The solutions, at room temperature, were purged with nitrogen previously to each experiment and kept under nitrogen atmosphere during the experiments. Both the sample and the electrolyte (Fluka, electrochemical grade) were dried under vacuum for several hours prior to the experiment.

### 3.4. Stability in Aqueous Medium and Air

The stability of all the complexes in water was evaluated by ^31^P NMR and UV–visible spectroscopies. Generally, a 5 mm NMR tube was charged in the air with the 10 mg of each complex and D2O (0.8 mL). ^1^H NMR and ^31^P NMR were monitored for four days to evaluate if there was any decomposition product. Decomposition/instability of the compound can be assessed by the disappearance of the NMR signals and/or by the appearance of new signals that could not be assigned to the known complex, beyond the appearance of free *m*TPPMS phosphine signals. Any eventual changes in the charge transfer bands between ruthenium and ligands were followed by UV–visible in the 300–900 nm range.

### 3.5. Octanol-Water Partition Coefficients

The lipophilicity of the complexes was measured by the shake-flask method [52]. Distilled water and *n*-octanol were mixed vigorously for 24 h at 25 °C, to promote solvent saturation in both phases, before the experiments. The phases were separated, and the compounds were dissolved in aqueous phase (≈10^−3^ M). Aliquots of stock solutions were equilibrated with octanol for 4 h in a mechanical shaker. The phase ratio was 2 mL/2 mL (water/*n*-octanol). The aqueous and octanol layers were carefully separated (by centrifugation at 5000 rpm for 10 min), and UV–vis absorption spectra of the compounds were registered in both phases. The concentration for each sample was determined using the calibration curve. Triplicate experiments have been performed for each complex and the averages were calculated.

### 3.6. In Vitro Anticancer Activity

The cytotoxic activity of all complexes was screened against A2780 (ovarian, Sigma-Aldrich, Lisboa, Portugal), MDAMB231 (breast, triple negative, ATCC), and HT29 (colon, ATCC) human cancer cells. Cells were grown in RPMI 1640 (A2780) or DMEM containing GlutaMax-I (MDAMB231), or McCoy’s (HT29) supplemented with 10% FBS and 1% penicillin/streptomycin (Invitrogen, Lisboa, Portugal). All cell lines were kept in a CO_2_ incubator (Heraeus, Hanau, Germany) with 5% CO_2_ at 37 °C in a humidified atmosphere. Cell viability was measured using the colorimetric MTT (3-(4,5-dimethylthiazol-2-yl)-2,5-diphenyltetrazolium bromide) assay [53]. For the assays, cells were seeded in 200 μL of complete medium in 96-well plates and incubated at 37 °C for 24 h prior to complex treatment to allow cell adherence. The stock solutions in water (20 mM) of the complexes were freshly prepared and used for sequential dilutions in medium within the concentration range of 10^−10^–10^−4^ M. Cisplatin was also included in this study as a positive control. After careful removal of the medium, 200 mL of a serial dilution of compounds in fresh medium was added to the cells, and incubation was carried out for 72 h at 37 °C. At the end of the treatment, the medium was discarded, and the cells were incubated with 200 mL of an MTT solution in PBS (0.5 mg mL^−1^). After 3 h incubation, medium was removed, and DMSO was added to solubilize the purple formazan crystals formed. The absorbance at 570 nm was measured using a plate spectrophotometer (Power Wave Xs, Bio-Tek, Winooski, VT, USA). Each experiment was repeated at least three times and each concentration tested in at least six replicates. The IC_50_ values were calculated from dose–response curves analyzed with the GraphPad Prism software (version 5.0).

### 3.7. Preparations of the Stock Solutions for Fluorescence Spectroscopic Measurements

Human serum albumin (HSA) was dissolved in 10 mM HEPES buffer (pH 7.4). The HSA concentration was determined spectrophotometrically using the absorbance value at 280 nm (ε_280_ = 36,500 M^−1^cm^−1^) [54]. Complex **6** was dissolved in 10 mM HEPES buffer (pH 7.4). A series of ruthenium-protein batch solutions were prepared by adding different concentrations of ruthenium solutions to the protein solution prepared previously. For fluorescence acquisition, the final HSA concentration was 2.5 μM, and the ruthenium were 0, 0.6, 1.2, 1.8, 2.5, 5, 7.5, 10, 12.5, 15, 17.5, 20, 22.5, 25, 27.5, 30, 32.5, 35, 37.5, 40, 42.5, and 45 μM. The solutions were stirred to ensure the formation of a homogeneous solution and stood in an incubator at 310.15 K for 24 h to stabilize and enhance the interaction time. The reference solutions were prepared following the procedures described above without protein.

### 3.8. Fluorescence Spectroscopic Measurements

Steady-state fluorescence measurements were carried out with a Fluorolog Model-3.22 spectrofluorometer from Horiba Jobin Yvon at 293.15, 298.15, and 310.15 K. All measurements were performed in Hellma^®^ semi-micro fluorescence cuvettes (Suprasil^®^ quartz, path length 10 × 4 mm, chamber volume 1.4 μL) with the 10 mm path length for the excitation of the sample. The excitation and emission slit widths were fixed at 4.0 nm, and the excitation wavelength was set at 295 nm to selectively excite the tryptophan 214 residue. The emission spectra were recorded from 305 to 550 nm. Solutions of ruthenium complex in 10 mM HEPES buffer pH 7.4 in corresponding concentrations were used as reference for the measured fluorescence spectra of protein-complex mixtures. No intrinsic fluorescence was displayed by complex **6** under our experimental conditions, and therefore, there was not any contribution to the Trp-214 fluorescence of HSA. The fluorescence intensities were corrected for the absorption of the exciting light and re-absorption of the emitted light to decrease the inner filter effect [55,56] using UV–visible absorption data recorded for each sample on a Jasco V-660 spectrophotometer in the range of 260 to 900 nm with 1 cm path quartz cells.

### 3.9. Site-Marker Competitive Studies

Competitive binding studies were carried out by fluorescence using the classical site marker warfarin. The concentrations of HSA and warfarin were kept equimolar. The addition of complex **6** to {HSA-warfarin} solution was done according to the procedure described above for the steady-state fluorescence quenching studies.

## 4. Conclusions

A series of new ruthenium(II)-cyclopentadienyl-based complexes containing water-soluble phosphine were developed and investigated as potential anticancer compounds. All the complexes are highly soluble in water, and, as expected, in general, the solubility depends on the number of phosphine ligands.

Cyclic voltammetric measurements clearly show the influence of the water-soluble phosphine coligand on the electronic environment and consequently on the redox behavior of piano-stool Ru(II) complexes. The compounds are relatively stable to oxidation even in aqueous solution, as demonstrated by the position of the anodic potential of the Ru(II)/Ru(III) couple in the range of 0.77 V to 1.17 V. In general, compounds with bidentate heteroaromatic ligands (**6**–**8**) are much more stable than compounds with monodentate ligands. Complexes with imidazole-based ligands have been shown to undergo hydrolysis after a few hours, the exception being complex **4**, which is stable for 24 h. The estimated log Po/w values are high, which indicates that the compounds studied are very lipophilic, suggesting a poor ability to cross the cell membrane. All the compounds exhibited high to moderate cytotoxicity against A2870, MDAMB231, and HT29 cells, the A2780 cells being considerably more sensitive to the compounds. Compound **7** was not cytotoxic for MDAMB231 and HT29 cells. From the overall results, it is not possible to correlate the cytotoxicity, number of phosphines, and lipophilicity. All the compounds proved to be less effective against the chemoresistant MDAMB231 and HT29 tumor cells.

The molecular interaction between complex **6** and HSA and HSA^faf^ was investigated by fluorescence spectroscopy. The results showed that the ruthenium complex binds strongly to these two albumin variants, the binding to HSA^faf^ being much stronger than the binding to HSA. The fact that binding is favored when fatty acids are not present could indicate that complex **6** can also compete with fatty acids for binding sites on HSA. In addition, the quenching observed upon the addition of the ruthenium compound to {HSA-warfarin} suggests that this complex binds to the protein in site I/subdomain IIA near the Trp-214.

## Data Availability

Not applicable.

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
