# Peer review of "Design and Anticancer Properties of New Water-Soluble Ruthenium–Cyclopentadienyl Complexes"

_pharmaceuticals, 2022, doi:10.3390/ph15070862_

Round 1

Reviewer 1 Report

Here Morais and co-workers report the synthesis and characterization of a number of new ruthenium complexes and their antiproliferative properties on several cancer cell lines.  Since I am an organic/organometallic chemist I can evaluate the synthesis and spectroscopic characterization part of this manuscript but not the cell line screens.

The new ruthenium complexes reported are well characterized by several spectroscopic techniques and exact mass.  In addition to this characterization the complexes were tested for water solubility via octanol water partition coefficients and then checked for their stability in air in water by UV-Vis and 31P NMR.

Assuming the cell line screens are well done I would recommend publication.

Typos/comments

In a number of places in this manuscript the authors use phosphine and phosphane to refer to the ligands in these complexes.  I think phosphane is used to refer to PH3 typically so I would encourage them to be consistent and just use phosphine to refer to the ligands used here.

line 16 complexes are emerging

line 17 their limited solubility

line 18 but In view and say To contribute...

line 20 butyl not butil

line 23 thiophene

line 74 I would define mTPPMS the first time you use it

line 86 butyl

line 110 already been observed

lines 121, 123, 125, 184, 203, 234 maybe more? I would change phosphane to phosphine

Author Response

Here Morais and co-workers report the synthesis and characterization of a number of new ruthenium complexes and their antiproliferative properties on several cancer cell lines.  Since I am an organic/organometallic chemist I can evaluate the synthesis and spectroscopic characterization part of this manuscript but not the cell line screens.

The new ruthenium complexes reported are well characterized by several spectroscopic techniques and exact mass.  In addition to this characterization the complexes were tested for water solubility via octanol water partition coefficients and then checked for their stability in air in water by UV-Vis and 31P NMR.

Assuming the cell line screens are well done I would recommend publication.

Typos/comments

1) In a number of places in this manuscript the authors use phosphine and phosphane to refer to the ligands in these complexes.  I think phosphane is used to refer to PH3 typically so I would encourage them to be consistent and just use phosphine to refer to the ligands used here.

Answer: We thank the reviewer for alerting us to the fact that we are using both names; as suggested by the reviewer, we adopt phophine throughout the manuscript.

2) line 16 complexes are emerging

Answer: We add “are” to the sentence.

3) line 17 their limited solubility

Answer: Corrected according to reviewer's suggestion.

4) line 18 but In view and say To contribute...

Answer: Corrected according to reviewer's suggestion.

5) line 20 butyl not butil

Answer: Corrected according to reviewer's suggestion.

6) line 23 thiophene

Answer: Corrected according to reviewer's suggestion.

7) line 74 I would define mTPPMS the first time you use it

Answer: We defined mTPPMS on line 74 according to the reviewer's suggestion and also in the abstract.

8) line 86 butyl

Answer: Corrected according to reviewer's suggestion.

9) line 110 already been observed

Answer: Corrected according to reviewer's suggestion.

10) lines 121, 123, 125, 184, 203, 234 maybe more? I would change phosphane to phosphine

Answer: Throughout all the manuscript we replace phosphane for phosphine.

Reviewer 2 Report

The authors Tania S. Morais and co-workers submitted the manuscript entitled „Design and anticancer properties of new water-soluble ruthenium-cyclopentadienyl complexes” to the MDPI-journal “Pharmaceuticals” in order to be considered for publication as an “Article”. Unfortunately, I did not find any supplemenatry information in the submission.

The manuscripts reports on the chemical synthesis of eight ruthenium(II)-cyclopentadienyl based complexes with the general formula [RuCp(mTPPMS)n(L)][CF3SO3] and the comprehensive analytical characterization (FT-IR, 1H, 13C and 31P NMR, UV-Vis, CV, MS). The water-solubity of the complexes as well as their (limited) stability in aqueous media was studies. In addition, the biological activity against three tumour cell lines, i.e. ovarian A2780, breast MDAMB231 and colon HT29 were investigated. Finally, the authors focused on the binding of complex 6 to HAS and fatty acid-free HAS, deducing a potential role as vehicle for the transport in plasma.

Please find below my comments on the manuscript:

“In this regard, the ruthenium complexes have emerged as one of the leading candidates for alternative platinum drugs [2–9]” I am not sure if citing all these references 2-9 is appropriate in this context as ref 4 reports on Pd complexes, ref 6 of Ir compounds and ref 7 on copper complexes.

“Among them, some ruthenium compounds revealed outstanding in vivo and in vitro activities, namely…” Can the authors consider to provide the chemical structures of these complexes in a figure for illustration of the introduction?

“Nevertheless, TM85 gained…” I think “TM85” has to be introduced prior to usage of this abbreviation. Also abbreviation “SCE” in Table 2.

Figure 1 should actually be Scheme 1. It is a bit pixelated and would therefore benefit from an improvement in resolution. The authors are asked to optimize the display. Please revise the caption by removing “Scheme Caption”.

Can the authors please judge in the manuscript on their selection of the particular ligands.

“Figure 2 shows the cyclic voltammograms of a representative example…” Can the authors kindly provide the cyclic voltammograms of the other complexes as supplementary material?

Table 4: Please add: The data represent the mean ± SD/SE of n=? independent experiments. (also Table 5)

The authors found that only complexes 6-8 (complexes with bidendate ligands) are stable under aqueous conditions for at least 4 days (i.e. 96 h), while complexes 1-5 were stable less than 1 day (i.e. 24 h). Then, the authors performed an MTT assay in aqueous cell culture medium with an incubation period of 72 h. Therefore, it is not possible to assign IC50 values to the compounds being not stable, since the anticancer effect could also be caused by degradation products. To properly assign the IC50 values to the complexes, it must be confirmed that the bunch of decomposition products is not biologically active. This would require figuring out all the degradation products and then excluding the activity of each of them.  What is actually added to this is that in the case of compound degradation in aqueous medium, the incubation time of the actual compound is significantly shorter, depending on the stability. A direct comparison with the other complexes (different or longer stability, development of the effect during the entire 72 h incubation) is therefore very difficult.

I also wonder about the precipitation of complex 8. If it precipitates, there is no solubility and the IC50 values of cytotoxicity should be interpreted with caution.

In their introduction, the authors talk about Ru-complexes as an alternative to Pt-complexes such as CisPt. One problem is the occurrence of resistance or cross-resistance to Pt-complexes. It would therefore be interesting to find out whether the water-soluble Ru-complexes are perhaps also active in a cell line that is resistant to CisPt. In the literature, a CisPt-resistant A2780 cell line is often used for this purpose. The authors are asked to investigate this aspect and thus increase the significance of their complexes.

Table 4: The IC50 values of CisPt against MDAMB231 and HT29 seem rather high, probably too high? In my opinion, there are rather data that suggest lower values, which of course makes the Ru-complexes presented here look worse.

The authors found, at least for complex 6 that it binds to HSA. In view of a potential mode of anticancer action it would be nice to know if this complex also binds to the DNA. Generally, information about the mechanism of action as evidenced by experiments would strengthen the impact of the manuscript. The authors are kindly aske to revise.

The authors are asked to improve formal aspects of the manuscript, such as: Lines 127-131: consistent use of space with cm-1, also Table 3 with “°C”, Figure 3b/line 258 /492/507/564 with “h”, line 506,509,561;  Line 132: N,N’ in italics? Also Line 229: n-octanol, Line 153/260: use of bold (6), Table 2-Compound: please unify the style (also Line 192), Lines 170/190/218: superscript, Line 565 subscript, typos line 269: Sine HAS has…, line 280: binding. This list is by no means complete, but is intended to help authors assess what kind of formal errors are still present in their manuscript and which should be corrected.

Author Response

The authors Tania S. Morais and co-workers submitted the manuscript entitled „Design and anticancer properties of new water-soluble ruthenium-cyclopentadienyl complexes” to the MDPI-journal “Pharmaceuticals” in order to be considered for publication as an “Article”. Unfortunately, I did not find any supplemenatry information in the submission.

Answer: We regret that you were unable to access the supplementary material.

The manuscripts reports on the chemical synthesis of eight ruthenium(II)-cyclopentadienyl based complexes with the general formula [RuCp(mTPPMS)n(L)][CF3SO3] and the comprehensive analytical characterization (FT-IR, 1H, 13C and 31P NMR, UV-Vis, CV, MS). The water-solubity of the complexes as well as their (limited) stability in aqueous media was studies. In addition, the biological activity against three tumour cell lines, i.e. ovarian A2780, breast MDAMB231 and colon HT29 were investigated. Finally, the authors focused on the binding of complex 6 to HAS and fatty acid-free HAS, deducing a potential role as vehicle for the transport in plasma.

Please find below my comments on the manuscript:

1) “In this regard, the ruthenium complexes have emerged as one of the leading candidates for alternative platinum drugs [2–9]” I am not sure if citing all these references 2-9 is appropriate in this context as ref 4 reports on Pd complexes, ref 6 of Ir compounds and ref 7 on copper complexes.

Answer: We thank the referee for pointing this error. In fact, references 2-9 are from the previous sentence. We already added the correct references.

2)“Among them, some ruthenium compounds revealed outstanding in vivo and in vitro activities, namely…” Can the authors consider to provide the chemical structures of these complexes in a figure for illustration of the introduction?

Answer: As suggested by the reviewer we added the structures of the complexes (new Figure 1).

3)“Nevertheless, TM85 gained…” I think “TM85” has to be introduced prior to usage of this abbreviation. Also abbreviation “SCE” in Table 2.

Answer: We thank the reviewer for alerting us to this point, so we replace the sentence:

“and being aware that the limited aqueous solubility of metallodrugs is one of the major limitations to its possible clinical application, we designed and synthesized a new molecular structure analogous to our lead RuCp(2,2’bipy)(PPh3)][CF3SO3] (TM34) that showed interesting in vitro and in vivo antitumor properties[38,40].”

by the new one, in which we present the structure of the TM85:

“and being aware that the limited aqueous solubility of metallodrugs is one of the major limitations to its possible clinical application, we designed and synthesized a new molecular structure [RuCp(2,2’bipy)(mTPPMSNa)][CF3SO3] (TM85) analogous to our lead TM34 that showed interesting in vitro and in vivo antitumor properties[38,40].”

We also introduced the definition of SCE in the Table 2 caption.

4)Figure 1 should actually be Scheme 1. It is a bit pixelated and would therefore benefit from an improvement in resolution. The authors are asked to optimize the display. Please revise the caption by removing “Scheme Caption”.

Answer: We agree that the figure must be a scheme. And we also exchanged the figure for one with higher quality.

4) Can the authors please judge in the manuscript on their selection of the particular ligands.

Answer: We include the next sentence to explain the choice of these ligands:

“The choice of these ligands was based on the promising results previously achieved with for the analogous complexes containing the {RuCp(PPh3)} fragment[36].”

5)“Figure 2 shows the cyclic voltammograms of a representative example…” Can the authors kindly provide the cyclic voltammograms of the other complexes as supplementary material?

Answer: As requested by the reviewer, all the other cyclic voltammograms were provided as supplementary material (Figures S1 to S9).

6)Table 4: Please add: The data represent the mean ± SD/SE of n=? independent experiments. (also Table 5)

Answer: We added the information requested in the legends of the two tables

7) The authors found that only complexes 6-8 (complexes with bidendate ligands) are stable under aqueous conditions for at least 4 days (i.e. 96 h), while complexes 1-5 were stable less than 1 day (i.e. 24 h). Then, the authors performed an MTT assay in aqueous cell culture medium with an incubation period of 72 h. Therefore, it is not possible to assign IC50 values to the compounds being not stable, since the anticancer effect could also be caused by degradation products. To properly assign the IC50 values to the complexes, it must be confirmed that the bunch of decomposition products is not biologically active. This would require figuring out all the degradation products and then excluding the activity of each of them.  What is actually added to this is that in the case of compound degradation in aqueous medium, the incubation time of the actual compound is significantly shorter, depending on the stability. A direct comparison with the other complexes (different or longer stability, development of the effect during the entire 72 h incubation) is therefore very difficult.

Answer: We agree with the criticism of the reviewer and we decided to remove all cytotoxicity data from the complexes that were not stable. We changed the text to explain why we only evaluated 3 compounds:

“The cytotoxic activity of the [Ru(h5-C5H5)(mTPPMS)(L)] [CF3SO3] complexes (6 – 8) was assessed in the ovarian A2780, triple negative breast MDAMB231 and colon HT29 cancer cells lines by the colorimetric MTT assay after 72 h exposure to different concentrations of the complexes 6 - 8. The cytotoxicity of the remaining compounds was not deter-mined due to the instability of the complexes.”

8) I also wonder about the precipitation of complex 8. If it precipitates, there is no solubility and the IC50 values of cytotoxicity should be interpreted with caution.

Answer: Prior to any evaluation we determined the solubility of the complexes (data on Table 3). In fact, the complex 8 precipitates at the concentrations we need to use to assess the stability of the compound. However, at the concentrations (much lower) used to determine the IC50 values the compound does not precipitate.

9)In their introduction, the authors talk about Ru-complexes as an alternative to Pt-complexes such as CisPt. One problem is the occurrence of resistance or cross-resistance to Pt-complexes. It would therefore be interesting to find out whether the water-soluble Ru-complexes are perhaps also active in a cell line that is resistant to CisPt. In the literature, a CisPt-resistant A2780 cell line is often used for this purpose. The authors are asked to investigate this aspect and thus increase the significance of their complexes.

Answer: During last years, we have been dedicated to developing families of complexes containing the RuCp fragment. We know that the mechanism of action of these compounds is different from that of CisPt, and that RuCp compounds are active in the A2780CisR cell line. For this reason, we do not consider necessary to assess cytotoxicity in this line. In this work, we intend to evaluate the potential of the water-soluble complexes and carry out a screening to choose the most promising ones to follow for further studies, which is why we chose such different lines. For the most promising, all studies related to its mechanism of action will be carried out in the future.

10) Table 4: The IC50 values of CisPt against MDAMB231 and HT29 seem rather high, probably too high? In my opinion, there are rather data that suggest lower values, which of course makes the Ru-complexes presented here look worse.

Answer: In fact, very different values are reported for cisplatin, but this is due to the fact that the cells undergo changes with each growth cycle. We trust in these values that we present here, some of them even determined in our laboratories.

11)The authors found, at least for complex 6 that it binds to HSA. In view of a potential mode of anticancer action it would be nice to know if this complex also binds to the DNA. Generally, information about the mechanism of action as evidenced by experiments would strengthen the impact of the manuscript. The authors are kindly aske to revise.

Answer: As we already mentioned, during last years, we have been dedicated to developing families of complexes containing the RuCp fragment. At this point, we already know some points of its mechanism of action, and we know that none of the RuCp complexes reach the nucleus, so the DNA will not be a target. Including the water-soluble ruthenium compound TM85 [ref. 40 -. J. Inorg. Biochem. 2014, 130, 1–14, doi: 10.1016/J.JINORGBIO.2013.09.013].

12)The authors are asked to improve formal aspects of the manuscript, such as:

Lines 127-131: consistent use of space with cm-1, also Table 3 with “°C”, Figure 3b/line 258 /492/507/564 with “h”, line 506,509,561; Line 132: N,N’ in italics? Also Line 229: n-octanol, Line 153/260: use of bold (6), Table 2-Compound: please unify the style (also Line 192), Lines 170/190/218: superscript, Line 565 subscript, typos line 269: Sine HAS has…, line 280: binding. This list is by no means complete, but is intended to help authors assess what kind of formal errors are still present in their manuscript and which should be corrected.

Answer: We appreciate the indication of all these typos that we have already corrected.

Round 2

Reviewer 2 Report

The authors submitted a revised version of their manuscript. They addressed every concern mentioned before and corrected errors, added supplementary data or provided rebuttal in the case they did not perform additional experiments. The manuscript fits the Special Issue "Metal-Based Agents in Drug Discovery" of the section "Medicinal Chemistry". I support futher processing of the manuscript. All the best!

Author Response

Thank you